# Usefulness of ${\dot{V}O}_{2}$ Kinetics and Biomechanical Parameters as Predictors of Athlete’s Performance in 800 m Running Race

**DOI:** 10.3390/sports11010015

**Published:** 2023-01-09

**Authors:** Vicente Torres Navarro, Jose Vicente Sánchez-Alarcos Díaz-Pintado, Diego Warr di Piero, Florentino Huertas Olmedo

**Affiliations:** 1Doctorate School, Catholic University of Valencia “San Vicente Martyr”, 46008 Valencia, Spain; 2Faculty of Physical Education and Sport Sciences, Catholic University of Valencia “San Vicente Martyr”, 46900 Torrent, Spain

**Keywords:** oxygen uptake, kinetics, kinematics, running, physiological response

## Abstract

Incremental tests to exhaustion have been usually employed as the “gold standard” to establish the fitness level of athletes. However, during real competition in many sport disciplines, exertion is not characterized by an increasing effort until failure. The purpose of this preliminary study was to add new evidence regarding the usability of parameters obtained from an on-field testing in 800 m running athletes. ${\dot{V}O}_{2}$ kinetics (mean, amplitude, phase time, and phase start time) and biomechanical parameters (velocity, stride frequency, and stride length) were analyzed in eight athletes during a maximal 800 m running race test. Our results showed that only the peak of blood lactate concentration after the 800 m test was correlated with the race time (*p* = 0.047). The race time was positively associated with both the phase duration and phase start time (all *p*-values < 0.05). Conversely, race time was negatively correlated with velocity, stride frequency, and amplitude (*p*-values < 0.05). Our results reveal that jointly studying the ${\dot{V}O}_{2}$ kinetics and biomechanical parameters during a maximal 800 m running race test is a useful tool to predict the athlete’s upcoming performance and improve the planning and control of the training process of 800 m running athletes.

## 1. Introduction

Traditionally, different types of incremental tests to exhaustion, mostly performed in the laboratory, have been used as the “gold standard” to establish the fitness level of athletes in sports in which running velocity at maximal oxygen uptake (${v\dot{V}O}_{2\max}$), maximal oxygen uptake (${\dot{V}O}_{2\max}$), or peak oxygen uptake (${\dot{V}O}_{2\mathrm{peak}}$) play a critical role in reaching the best performance [1,2,3]. However, the athlete’s effort during real competition in these sport disciplines, such as the 800 m running race, is not characterized by a progressive effort to exhaustion. Accordingly, the incremental tests do not specifically reproduce the fluctuations in velocity that have a key role in the athlete’s physiological response during competition [4,5] or the regulation of the rate of energy expenditure [6]. Furthermore, most of laboratory tests also fail in contemplating biomechanical factors, such as stride frequency and stride length, which modulate the variation in running speed [7,8,9].

To understand the dynamic response of oxygen uptake (${\dot{V}O}_{2}$) while competing, it is essential to study the kinetics of ${\dot{V}O}_{2}$ during the transition between rest and exercise $({\dot{V}O}_{2}$ on-kinetics) [10,11]. Whipp and Ward [12] distinguished three phases that characterize ${\dot{V}O}_{2}$ kinetics: phase I (cardio dynamic component), phase II (primary or fast component), and phase III (slow component or steady state). These phases are defined by a different ${\dot{V}O}_{2}$ kinetic response according to the exercise intensity: moderate, heavy, severe, and extreme [13,14,15]. During severe and extreme exercise, such as the 800 m running race, Hanon and Thomas [16] described a phase IV (${\dot{V}O}_{2}$_decreases_), from the instant when ${\dot{V}O}_{2\mathrm{peak}}$ is reached to the end of the race. In predominantly extreme exercise bouts, the kinetic of the ${\dot{V}O}_{2}$ fast component is exponential, with not enough time for the phenomenon of the ${\dot{V}O}_{2}$ slow component (${\dot{V}O}_{2}$_sc_) [17] to appear and develop, sometimes meaning that the exercise ends before ${\dot{V}O}_{2\max}$ is reached [13,18]. Under these conditions of extreme physiological demand, some biomechanical parameters of running (stride frequency and length) are altered, causing a decrease in athletic performance [9].

Analysis of ${\dot{V}O}_{2}$ kinetics and its relationship with performance has recently gained popularity in other sports, such as swimming [19]. However, few studies have used this type of approach in athletics [20,21], and those did not analyze ${\dot{V}O}_{2}$ kinetics in concurrence with biomechanical parameters during the race. Limitations have been shown by different assessment methods predominantly used to estimate performance in events such as the 800 m running race; these assessment methods include mathematical modelling and accumulated oxygen deficit (AOD) on treadmills. Accordingly, it is important to study the usefulness of new and more ecological ${\dot{V}O}_{2}$ kinetics testing procedures, including the analysis of the athlete’s biomechanical (kinematic) behavior during the different phases of the race.

## 2. Materials and Methods

### 2.1. Participants

Eight 800 m runners (age: 25.00 ± 8.42 years; height: 1.77 ± 0.05 m; and weight: 65.13 ± 8.10 kg) participated in the study. Participants’ selection criteria were the following: being over 18 years of age, minimum experience of 2 years competing in official 800 m races (2.40 ± 0.32 years), training frequency of 3 or more sessions per week, and without interruption in their sports practice in the last 6 months. Athletes’ personal best (PB) ranged from 119.60 s to 143.26 s. All of them were informed of the purpose and the protocol of the study. They provided a written informed consent. The experimental procedure was approved by the University’s Ethics Committee (code UCV2017-2018-93).

### 2.2. Experimental Protocol

The participants completed two evaluation sessions: First—familiarization session, and second—800 m running race test. All participants performed the two sessions in the same order and at a similar time of the day (between 10.00 am and 1.00 pm), with a minimum recovery time of at least 48 h between sessions. Considering the dates of official competitions, individual testing was conducted approximately in the same period of the athlete’s annual planning, during the specific preparatory period (from 2 to 6 weeks before one of their main target competitions). This time corresponded approximately between the 12th and 18th week of their annual preparation (from 11 November to 22 December). All participants were informed of the recommendations to be followed during the 48 h preceding the testing sessions (abstaining from taking stimulant substances or performance enhancers, following the pre-competition diet, drinking sufficient fluids, and refraining from doing any intense or high-load training).

### 2.3. Instruments and Material

Physiological respiratory variables were collected and recorded using a portable Oxycon Mobile gas analyzer (Jaeger, Heidelberg, Germany), taking a sample every 5 s. The gas analyzer was automatically calibrated following the recommended protocols and the manufacturer’s instructions [22]. Peak blood lactate concentration ((La ^−^)_peak_) was measured with a Lactate Pro 2 analyzer (Arkray Inc., Kyoto, Japan), obtaining capillary blood samples from the ear lobe. Heart rate (HR) and running velocity were also recorded using a Forerunner^®^ 405 watch (Garmin, Olathe, KS, USA). Athletes’ stride length and frequency were measured using individual video analysis with a camera (Sony HDR-CX405, Sony Corporation, Tokyo, Japan) placed in the stadium control tower, with a sampling rate of 50 Hz. The kinematic variables were subsequently analyzed using Kinovea software (version 0.8.7), following previous recommendations [23]. Running velocity (Vr), stride frequency (SF), and stride length (SL) were calculated every 50 m. This was accomplished by placing 16 marks around the entire track: eight marks at each 50 m partial point, and eight different colored marks at different points to control the parallax effect that could occur when using a single camera.

#### 2.3.1. Session 1—Familiarization

The familiarization session allowed the athlete to get used to the portable gas analyzer. In this session, an interval training session while wearing the portable gas analyzer was performed on the running track (8 × 400 m/rec. 90 s) at 20–25% below the average velocity of PB.

#### 2.3.2. Session 2—800 m Running Race on Field Test

Each participant performed the 800 m running race on the athletics track simulating competition intensity, wearing the gas analyzer, GPS, and chest-strap HR monitor. Before the test, the participants performed a pre-competition warm-up [16,24]. After the warm-up, the athletes had a recovery time of 3 min [25]. The test started after an acoustic signal once the gas analyzer and GPS were synchronized. The athletes selected their own running pace during the test on the basis of their best performance in competition and their own experience. The athletes received information regarding their time at 400 m. Lactate samples were taken from the earlobe under resting conditions before the start of the warm-up and 1, 3, and 5 min immediately after completing the test, using the highest value ((La ^−^)_peak_) for our analyses. All the field-testing sessions were performed under similar environmental and weather conditions (25 m altitude, 22–26 °C temperature, and 45–50% relative humidity).

### 2.4. Data Analysis

The ${\dot{V}O}_{2}$ kinetics were divided into four phases: phase I (cardio dynamic component or CD), phase II (primary or fast component or P), phase III (slow component or SC), and phase IV (decrease or D). The transitions between phases were determined by the ${\dot{V}O}_{2}$ kinetic response. By calculating the mean ${\dot{V}O}_{2}$ value for each 50 m segment, the transition of each phase coincides with a multiple of that 50 m distance. Thus, the transition between the CD and P phases was established on the basis of the fulfilment of the following criteria, which are usually very close in time: (1) first point at which there was a sharp reduction in the increase in ${\dot{V}O}_{2}$ over time (first inflection) after the first major exponential increase in ${\dot{V}O}_{2}$ from the start of running [26,27,28] and (2) point at which there is a drop in the respiratory exchange ratio (RER) [27,28]. HR was used as a measure of secondary confirmatory criterion to those proposed for the determination of phase I (cardio dynamic component). The transition between phases P and SC was established on the detection of the “drift” or breakpoint of ${\dot{V}O}_{2}$ over time with an increase of ≥150 mL min^−1^, following a stabilization of the increase in ${\dot{V}O}_{2}$ that continues from the CD phase [29,30,31,32,33]. Finally, the transition between phases SC and D was defined by the point at which the athlete was no longer able to maintain the ${\dot{V}O}_{2}$ plateau over time, taking the first ${\dot{V}O}_{2}$ decrease breakpoint value of ≥150 mL min^−1^ as a reference, in line with the criteria described by Hill and Lupton [34].

For this ${\dot{V}O}_{2}$ kinetics analysis, and according to previous studies [11,16,35], the following ${\dot{V}O}_{2}$ kinetic parameters were calculated for each phase: the mean ${\dot{V}O}_{2}$ value (Ⴟ); the amplitude (Δ), defined as the difference in ${\dot{V}O}_{2}\;$value from the start to the end of the phase; the time constant (τ), defined as the duration of each phase; and the time delay (TD), defined as the time from the start of the running test to the start of each phase.

The kinematic parameters during the test were determined as follows. The Vr of each 50 m segment was calculated as the average of the velocity values recorded by the GPS during that segment. SF was defined as the number of foot landings made in the segment divided by the segment time. SL was determined as the average horizontal distance between the point of foot contact between two consecutive landings [36,37], and it was calculated as the segment distance divided by the number of landings in the segment. The number of landings in each 50 m segment was counted based on the previous standardization established by the study’s authors. Once this had been established, inter-rater reliability [38] was calculated using the intraclass correlation coefficient (ICC), for which very high values were reported (ICC = 0.996; 95% confidence interval (CI) = 0.992–0.998).

### 2.5. Statistical Analysis

Data are presented as mean and standard deviation (SD). The normal distribution and sphericity of all the variables were confirmed using the Shapiro–Wilk test and Mauchly’s test, respectively. One-way repeated measures ANOVAs were carried out to identify differences in dependent variables among the four running phases of the 800 m race. When significant effects were observed in variables with more than two levels, paired t-tests were performed, applying the Bonferroni correction. Pearson correlation coefficients were used to analyze the relationship between the various study variables and running performance (time obtained in the 800 m race). Effect size was reported using partial eta squared (η_p_^2^).

Sample size of the presenter study (*n* = 8) is similar to previous research on this topic [16,20,39]. Nevertheless, a sensitivity analysis using the G*Power 3 [40] showed that in a repeated measures ANOVAs, the minimum effect size that could be detected (for α = 0.5 (two-tailed) and 1 − β = 0.80 for 4 groups) is f = 0.791. For a Pearson correlation coefficient, our sample would allow us to sense effects of r = 0.821 for α = 0.5 (two-tailed) and 1 − β = 0.80. The level of statistical significance was set at *p* ≤ 0.05. Statistical procedures were carried out using SPSS software, version 21.0 (SPSS Inc., Chicago, IL, USA).

## 3. Results

Table 1 displays the values obtained for the variables analyzed in the 800 m running race test.

Our results show that the athletes’ performances during the 800 m running race test were worse than the participants’ PBs obtained in official competition. When analyzing the relationship among all the studied variables and the time obtained in the 800 m test, only the (La ^−^)_peak_ value showed a statistically significant negative correlation (*r* = −0.714, *p* = 0.047).

### 3.1. ${\dot{V}O}_{2}$ Kinetics

Table 2 depicts the descriptive results for the physiological parameters analyzed in the different phases of the evolution of the ${\dot{V}O}_{2}$ during the test (see Figure 1 for an example of athletes’ ${\dot{V}O}_{2}$ parameters evolution). Repeated measures ANOVA showed statistically significant differences between phases in the mean values of ${\dot{V}O}_{2}$ (*F*_(3,5)_ = 76.57; *p* = 0.000; η_p_^2^ = 0.916). Post hoc analyses showed that ${\dot{V}O}_{2}$ increased significantly, by 138.82% from the start of the test to the P phase (*p* = 0.0001), and by 23.57% from the P to the SC phase (*p* = 0.007), with a 7.19% nonsignificant decrease observed in the final phase of the race, in phases SC and D (*p* = 0.015). Furthermore, our results showed statistically significant changes among phases in the curve amplitude values (*F*_(3,5)_ = 60.97; *p* = 0.0001; η_p_^2^ = 0.897). This amplitude increased from the beginning of the test up to the P phase (*p* = 0.011) and from this phase onwards; the increase was smaller in the SC phase (*p* = 0.0001) and decreased from the beginning of the D phase to the end of the test.

Our analyses show that neither *Ⴟ* nor Δ were significantly related to the final time achieved in the test (*p*-values > 0.338). Nevertheless, significant positive correlations were observed between the performance obtained in the test and different parameters associated with the duration of the phases: *τ_P_* (*r* = 0.899, *p* = 0.002), *τ_SC_* (*r* = 0.913, *p* = 0.002), and *τ_D_* (*r* = 0.794, *p* = 0.019), as well as with the start time of the phases from the beginning of the test TD_SC_ (*r* = 0.886, *p* = 0.003) and TD_D_ (*r* = 0.989, *p* = 0.0001).

### 3.2. Evolution of Biomechanical Parameters during the Test

Table 3 shows the descriptive results for Vr, SF, and SL in each of the phases of the 800 m test.

Repeated measures ANOVAs showed statistically significant changes among phases in SF (*F*_(3,5)_ = 10.71, *p* = 0.013, ηp2 = 0.865) and SL (*F*_(3,5)_ = 14.76, *p* = 0.006, η_p_^2^ = 0.899), while no differences were found in Vr (all *p*-values > 0.293). Post hoc analyses showed a significant reduction in SF from CD to P phase (*p* = 0.005), remaining unchanged from P until the end of the race (*p* > 0.05). However, unlike SF, SL varied significantly throughout the test, increasing by 7.19% from the beginning to the P phase (*p* = 0.002) and remaining significantly unchanged in the last three phases of the race (P, SC, and D, *p*-values > 0.654).

Statistically significant negative correlations were observed between Vr and final performance in the P phase (*r* = −0.781, *p* = 0.022), SC phase (*r* = −0.852, *p* = 0.007), and D phase (*r* = −0.791, *p* = 0.019). On the other hand, SF was negatively related only to race time in the P phase (*r* = −0.781, *p* = 0.022), SL in the P phase (*r* = −0.753, *p* = 0.031), and SC (*r* = −0.869, *p* = 0.005) (Figure 2).

## 4. Discussion

The purpose of the present study was to describe the usefulness of jointly measuring ${\dot{V}O}_{2}$ kinetics and biomechanical parameters during different stages of an 800 m running race as predictors of an athlete’s performance. Here, we have described the ${\dot{V}O}_{2}$ kinetic response to exercise and its relationship with specific kinematic parameters in this athletic discipline using a non-invasive and more specific way, simulating competitive conditions. Until now, these aspects had been studied with a similar methodology only in middle-distance swimmers [11,19] with the purpose of being used in the control and quantification of the training plan and the prescription of individual workload [41]. Our study is the first investigation addressing these issues in the 800 m running race.

Performance in an 800 m running race is modulated by the mixed contribution of both the aerobic and anaerobic systems, as confirmed by some of the analyzed physiological parameters described in our results. Our findings showed lower ${\dot{V}O}_{2\mathrm{peak}}$ and (La^−^)_peak_ values than those reported in previous studies [16]. These differences could be justified by the level of the participants (regional level) and the performance (race time) obtained in the test, approximately 94.17 ± 3.94% of their PB. Underestimating the performance in non-ecological testing conditions is usual in sport performance research, and it is usually attributed to the fact that athletes must carry extra instruments (e.g., gas analyzer, pulsometer, etc.) with the disturbances and alteration of normal conditions under which they compete. Moreover, the absence of opponents (our test was an individual time trial without other contestants and involved self-regulated running pace focusing on achieving optimal individual performance) and lack of relevance of the time achieved, could affect the motivational and emotional state necessary to do their best during testing conditions [42,43]. Further research should include these environmental issues to investigate their influence on performance and the other dependent variables. In any case, our research did not aim to compare performance in different environmental conditions or time but to describe the usefulness of this mixed specific methodology of evaluation of predictors of performance in conditions of maximum demand. Although the level of the athletes was lower than in previous studies and they did not achieve their best performance, this does not affect the final purpose of the study.

Regarding the relationship between athletes’ performance and physiological variables, our results showed that only (La ^−^)_peak_ values correlated negatively with the time achieved in the race. This result is in line with previous findings showing that high lactate production and tolerance are the most important adaptive processes influencing success in high-intensity events [44,45]. Our findings confirm that the use of lactate analysis in maximal field tests is a useful measure in the estimation of performance in 800 m athletes.

Concerning the ${\dot{V}O}_{2}$ dynamic response during our simulation of the 800 m running race, results showed that in the first ${\dot{V}O}_{2}$ transition, the τ_CD_ coincides with the time response lasting up to 20 s proposed by Whipp et al. [28], in line with previous findings observed in middle-distance swimmers [10,15]. Previous evidence confirmed that shortening the length of the CD phase leads to a rapid increase in ${\dot{V}O}_{2}$ and a shorter P phase, thus enabling athletic performance [26]. Our results, showing a positive correlation between the duration of P phase and the time achieved in the race test, add new evidence and can be applied to the athletic races, confirming that a shorter length of the P phase is associated with better athletic performance [10]. Regarding the duration of the SC phase, our results showed that it accounted for 42.53% of the total time in the 800 m race test. These values are similar to those obtained in previous studies with 400 m swimmers [10,19]. It should be noted that a faster ${\dot{V}O}_{2}$ kinetic response is associated with better performance in the test [10], and, therefore, shorter *τ_P_* and TD_SC_ are associated with greater tolerance to fatigue during exercise [13,46]. This association between phase duration and performance has been confirmed by our observed correlation between the time obtained in the 800 m test and the duration of the phases: *t_P_*, *τ_SC_*, and *τ_D_*, as well as with the start time of the phases from the beginning of the test. Our findings are in line with those described by Reis et al. [10], whose *τ_P_* kinetic parameter correlates significantly with the 400 m swimming time in both heavy and severe exercise. These results show that a shorter *τ_P_* is related to better performance, a very useful aspect for coaches to consider in their training plan. The relatively small sample size used in our study, as in most of the previous studies on this topic, is due to the difficulties in finding participants of similar performance level in this sport modality [11,16,39], and it could constrain the generalization of our results. Therefore, further investigations should replicate our findings by increasing the number of participants, as significant effects have been found that justify the interest of the proposed evaluation methodology.

More importantly for the purposes of our study, the concurrent study of ${\dot{V}O}_{2}$ kinetics and biomechanical parameters in athletic races was confirmed to be a useful methodological approach to improve the understanding of the running behavioral and physiological response. Our results have shown how the decrease in ${\dot{V}O}_{2\mathrm{peak}}$ at the end of the race is almost simultaneously paired with a decline in Vr in the D phase caused by a reduction in SF and SL. During the 800 m race, these changes at the end of the race have been attributed to the occurrence of peripheral muscle fatigue [9,20,47]. However, in our study, no significant differences were found among the P, SC, and D phases for any of the biomechanical variables studied.

These results are in discrepancy with those reported by previous research with elite athletes, showing a decrease in velocity in the final phase of the race [20]. This could be explained by the absence of opponents and competitive environment in our study. These conditions could lead the athletes to adopt a different race strategy (“positive pacing”), starting the race in a more controlled manner that leads to less metabolic acidosis at the end of the race [48,49], resulting in more stable biomechanical behavior up to the end of the race. However, given that these biomechanical parameters undergo changes during the training process [50], and that a “positive pacing” strategy is used during competition in the 800 m race with a “fast-start” [42,51,52], provoking an extremely fatiguing finish, it is necessary to analyze the biomechanical parameters together with the ${\dot{V}O}_{2}$ kinetics in the different phases of the race.

Our results confirm the need to differentiate the kinetics of VO2 observed during specific field tests vs. laboratory conditions, since the variables usually studied in a laboratory test (VO_2_, VO_2max_, and VO_2peak_), are registered as absolute values that do not correlate with the performance obtained in the race. Furthermore, the running protocol used in the laboratory tests, without any similarity to the real running race strategy, does not guarantee the validity of the VO2 kinetics analyses. Future lines of research include the comparison of field and laboratory studies to objectively confirm this fact.

The methodological approach for the athletic assessment used in this study represents a step forward that improves the knowledge on the definition of the ${\dot{V}O}_{2}$ response profile in 800 m athletes [20] jointly with biomechanical aspects. These issues are crucial, considering that the 800 m race requires the ability to coordinate neuromuscular/mechanical (SL and SF) and metabolic components to maintain the race pace efficiently [53]. Thus, we consider that the analysis of ${\dot{V}O}_{2}$ kinetics and biomechanical behavior in the different phases of specific field tests can help optimize individualized training according to the event and the athlete’s characteristics.

## 5. Conclusions

Our proposal represents an innovative methodology to estimate and predict the athlete’s performance in the 800 m running race. This proposal is a more ecological solution to analyze the ${\dot{V}O}_{2}$ kinetics combined with specific biomechanical factors under conditions similar to the real competition.

Better performance in the 800 m race is related primarily to faster ${\dot{V}O}_{2}$ kinetics. Considering the nature of this athletic modality and the fact that there are different types of runners with different physiological and biomechanical characteristics, our preliminary study represents a step forward in the methodology for evaluating and optimizing the individual training process, which will improve the knowledge regarding the ${\dot{V}O}_{2}$ response profile and biomechanical parameters according to the race strategy.

## Figures and Tables

**Figure 1 sports-11-00015-f001:**
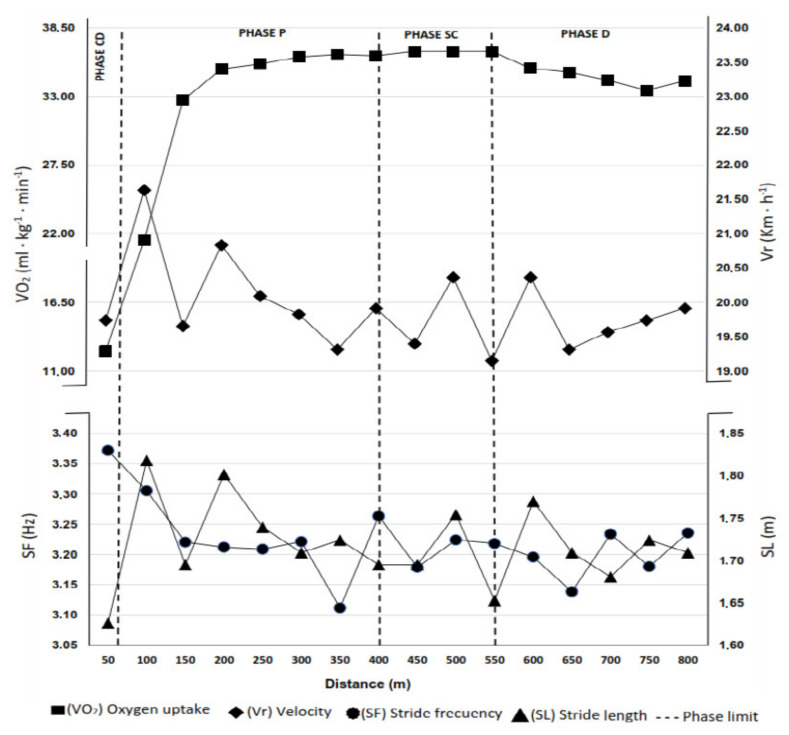
Evolution of the amplitude for ${\dot{V}O}_{2}$, velocity, stride frequency, and stride length of an athlete during the 800 m field test.

**Figure 2 sports-11-00015-f002:**
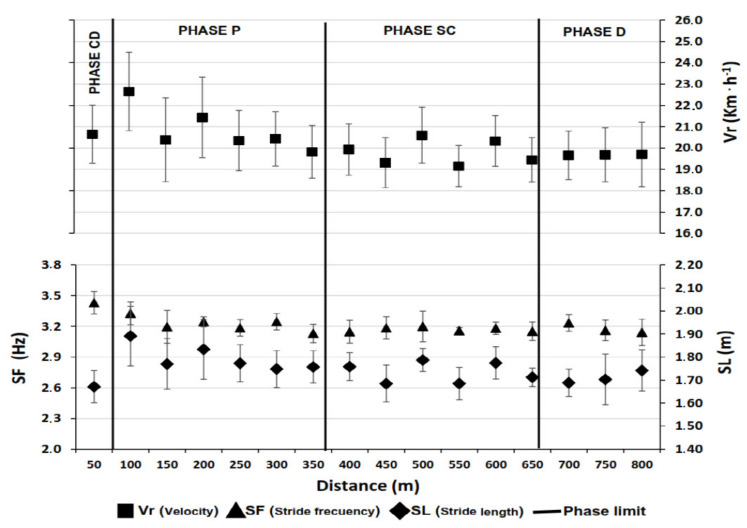
Evolution of the mean for velocity, stride frequency, and stride length of the athletes during the 800 m test. **Note:** Vertical bars represent Standard Deviation.

**Table 1 sports-11-00015-t001:** Athlete’s individual results obtained during 800 m running race test.

Participant	PB Time (s)	Test Race Time (s)	%PB	${\;\dot{V}O}_{2\mathrm{peak}}\;\;\left(mL\;\cdot \;{kg}^{-1}\;\cdot \;{min}^{-1}\right)$	(La ^−^)_peak_ (mmol L^−1^)
1	129.67	137.4	94.32	50.1	15.7
2	143.26	144.6	99.02	36.4	15.5
3	132.56	143.1	92.67	51.1	14.2
4	134.15	140.1	95.76	36.8	22
5	119.6	130.1	91.89	50.3	18.9
6	133.58	152.6	87.49	46.2	11
7	142.8	144.7	99.48	50.1	14.3
8	133.61	153.1	92.71	51.2	13.6
Mean ± SD	133.65 ± 7.47	143.24 ± 7.60	94.17 ± 3.94	46.52 ± 6.32	15.7 ± 3.39

**Note:** ${\dot{V}O}_{2\mathrm{peak}}$: Peak oxygen uptake; **(La ^−^)_peak_**: Peak blood lactate concentration; **PB**: Personal best; **%PB**: Personal best percentage obtained in test.

**Table 2 sports-11-00015-t002:** Athlete’s individual ${\dot{V}O}_{2}$ kinetics parameters in the 800 m running race test.

	*Ⴟ* (mL kg^−1^ min^−1^)				Δ (mL·kg^−1^ min^−1^)				*τ* (s)				TD (s)			
Phases	CD	P	SC	D	CD	P	SC	D	CD	P	SC	D	CD	P	SC	D
Participant																
1	19.47	44.60	54.65	49.97	14.37	32.97	3.38	6.02	7.88	46.75	55.16	27.68	-	7.88	54.63	109.79
2	12.60	32.92	36.04	34.02	9.54	23.75	−1.40	0.70	9.12	53.48	54.72	27.36	-	9.12	62.6	117.32
3	13.87	42.07	55.14	50.77	9.69	33.43	8.61	6.31	8.11	50.2	55.44	29.28	-	8.11	58.31	113.75
4	16.85	30.60	34.73	33.20	12.81	16.50	1.30	2.05	9.28	50.2	53.64	26.96	-	9.28	59.48	113.12
5	15.9	39.09	52.43	52.67	10.1	33.65	4.80	1.15	8.28	48.76	48.95	24.15	-	8.28	57.04	105.99
6	22.6	40.97	45.02	41.13	18.35	19.85	0.25	4.33	9.00	57.32	57.6	28.64	-	9.00	66.32	123.92
7	12.70	30.89	36.78	34.67	10.2	23.70	0.10	2.20	9.1	53.52	54.64	27.5	-	9.1	62.62	117.26
8	12.37	40.57	53.71	49.60	9.06	33.43	8.61	6.31	9.24	57.24	57.84	28.74	-	9.24	66.48	124.32
Mean ± SD	15.79 ± 3.69	37.71 ± 5.44 *	46. 06 ± 9.03 *^,**†**^	43.25 ± 8.40 *^,**Ŧ**^	11.76 ± 3.23	27.16 ± 7.02 *	3.20 ± 3.85 *^,**†**^	3.63 ± 2.38 *^,**†**^	8.75 ± 0.56	52.18 ± 3.86	54.74 ± 2.75	27.53 ± 1.58	-	8.75 ± 0.56	60.93 ± 4.29	115.68 ± 6.39

**Note: ***Ⴟ* mean ${\dot{V}O}_{2}$ values; **Δ**: amplitude of the curve; ***τ***: duration of the phase; **TD**: time delay. * Significant difference (*p* ≤ 0.05) with **CD**; **^†^** Significant difference (*p* ≤ 0.05) with **P**; **^Ŧ^** Significant difference (*p* ≤ 0.05) with **SC**.

**Table 3 sports-11-00015-t003:** Athlete’s individual values for mean velocity, stride frequency, and stride length during the different phases of the 800 m on field test.

	Vr (km h^−1^)				SF (Hz)				SL (m)			
Phases	CD	P	SC	D	CD	P	SC	D	CD	P	SC	D
Participant												
1	22.84	23.22	19.60	19.53	3.65	3.34	3.14	3.12	1.74	1.93	1.74	1.74
2	19.74	20.22	19.75	19.74	3.37	3.21	3.20	3.22	1.63	1.75	1.71	1.70
3	22.19	21.56	19.50	18.44	3.45	3.23	3.15	3.11	1.79	1.85	1.72	1.65
4	19.40	21.59	20.20	20.06	3.23	3.22	3.12	3.15	1.67	1.86	1.80	1.77
5	21.74	22.18	22.08	22.37	3.50	3.29	3.32	3.32	1.72	1.87	1.85	1.87
6	20.00	18.87	18.76	18.86	3.44	3.11	3.12	3.11	1.61	1.68	1.67	1.69
7	19.78	20.21	19.78	19.64	3.38	3.24	3.21	3.22	1.63	1.73	1.71	1.70
8	19.48	18.91	18.68	18.79	3.41	3.12	3.11	3.17	1.59	1.68	1.67	1.57
Mean ± SD	20.64 ± 1.37	20.84 ± 1.55	19.79 ± 1.57	19.67 ± 1.21	3.42 ± 0.11	3.22 ± 0.07 *	3.17 ± 0.07 *	3.17 ± 0.07 *	1.67 ± 0.07	1.79 ± 0.95 *	1.73 ± 0.62	1.71 ± 0.87

**Note: Vr**: running velocity; **SF**: stride frequency; **SL**: stride length. * Significant difference (*p* ≤ 0.01) with **CD**.

## Data Availability

Not applicable.

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
