# Peer review of "Usefulness of Kinetics and Biomechanical Parameters as Predictors of Athlete’s Performance in 800 m Running Race"

_sports, 2023, doi:10.3390/sports11010015_

Round 1
Reviewer 1 Report
First of all, I was very excited to evaluate this article and read it line by line. I thank the authors for designing and completing this study, the results of which are very important for the coaches and athletes in the field. The article is of very high scientific standard, but there are only minor changes and points that need to be corrected. comments are as follows
Intro as a whole is well designed and explianed why this study was needed.
Second paragrapgh last sentence: ….., reducing sports performance.. check the grammer,
Second paragrapgh; seperate here [V̇O2sc; 17].
I think the 3th and 4th paragrapgh can be combined
2.2. experimental protocol
Can you state whole study time period lasted between the dates… ?
3.1 vo2 kinetics; “Repeated measures ANOVA showed the between phases statistically significant differences in the mean V̇O2 values (F(3,5) = 76.57; p = .000; ηp2 = .916).” check the grammer.
Discussion was well organized.
Author Response
Dear Editor, Ms. Mara Pop
Assistant Editor, Sports.
Please find attached a Word copy of a revised version of the manuscript entitled ‘Usefulness of kinetics and biomechanical parameters as predictors of athlete´s performance in an 800 m running race’ by Vicente Torres et al., which we originally submitted for publication in Sports on November 26th, 2022.
On behalf of the authors, I would like to thank both the reviewers and you for providing such thorough and constructive feedback on our original submission. We certainly believe that the manuscript has improved from all the positive feedback and thoughtful comments generated during this review.
We have detailed our responses to both reviewer’s comments and suggestions on the pages that follow.
Given that we have made the changes recommended by yourself and the reviewers we hope that you will now find the enclosed manuscript suitable for publication in Sports.
Yours sincerely,
Vicente Torres on behalf of all authors
Reviewers' Comments to Author
Reviewer 1
On behalf of the authors, I would like to thank to the Reviewer 1 his/her positive feedback and constructive comments and suggestions on our original submission.
We apologize to you for the language mistakes. The manuscript has been revised again by a native English speaker with wide experience in translations for the field of Sport Physiology and Movement Sciences. We undoubtedly believe that the manuscript has been improved.
We have detailed our responses to the Reviewer 1 comments and suggestions on the pages that follow.
---
Comment 1
Second paragraph last sentence: …., reducing sports performance... check the grammar.
Modification (Line 115-117):
Thanks for the suggestion. We have modified the paragraph to “Under these conditions of extreme physiological demand, some biomechanical parameters of running (stride frequency and stride length) are altered, causing a decrease in athletic performance [9]”
---
Comment 2
Second paragraph; separate here [V̇O2sc; 17].
Modification (Line 114):
Done. The word “V̇O2sc” has been separated “V̇O2 sc”.
---
Comment 3
I think the 3rd and 4th paragraph can be combined.
Modification:
Done. Both paragraphs have been combined.
---
Comment 4
2.2. Experimental protocol. Can you state whole study time lasted between the dates…?
Modification (Line 144-148):
Thanks for your suggestion. We have explained clearer the “temporal window” when the testing protocol were done. We have included the text: “Considering the dates of official competitions, individual testing was done approximately in the same period of the athlete´s annual planning, during the specific preparatory period (from 2 to 6 weeks before one of their main target competitions). This time correspond approximately between the 12th and 18th week of their annual preparation (from November 11th to December 22nd)”.
---
Comment 5
3.1 vo2 kinetics; “Repeated measures ANOVA showed the between phases statistically significant differences in the mean V̇O2 values (F (3,5) = 76.57; p = .000; ηp2 = .916).” check the grammar.
Modification (Line 344-346):
Repeated measures ANOVA showed statistically significant differences between phases in the mean values of V̇O2 (F (3,5) = 76.57; p = .000; ηp2 = .916).
---
We thank Reviewer 1 for your detailed comments and suggestions to improve the quality of our previous submission.
Given that we have made the changes recommended by yourself, we hope that you will now find the enclosed manuscript suitable for publication in this journal.
Yours sincerely,
Vicente Torres on behalf of all authors

Reviewer 2 Report
The main objective of authors is measure VO2 kinetics and relate it with biomechanical parameters in order to predict 800m performance.
The study is well conducted and methods are clearly explained. While, in the discussion the authors explain in excess the effects of sample size in comparation to the the usefulness of jointly measuring V̇O2 kinetics and biomechanical parameters.
Minor changes
In order to see the point at which there is a drop in the respiratory exchange ratio (RER), it could be interesting to show evolution of V̇CO2, in figure 1
Correlation value of lactate is explained in abstract but later on it is not present in results, it is necessary to explain.
Authors comment in methods that Heart rate (HR) is measured, but there isn’t any reference about this variable in results or discussion, why results are not explained?
Author Response
Dear Editor, Ms. Mara Pop
Assistant Editor, Sports.
Please find attached a Word copy of a revised version of the manuscript entitled ‘Usefulness of kinetics and biomechanical parameters as predictors of athlete´s performance in an 800 m running race’ by Vicente Torres et al., which we originally submitted for publication in Sports on November 26th, 2022.
On behalf of the authors, I would like to thank both the reviewers and you for providing such thorough and constructive feedback on our original submission. We certainly believe that the manuscript has improved from all the positive feedback and thoughtful comments generated during this review.
We have detailed our responses to both reviewer’s comments and suggestions on the pages that follow.
Given that we have made the changes recommended by yourself and the reviewers we hope that you will now find the enclosed manuscript suitable for publication in Sports.
Yours sincerely,
Vicente Torres on behalf of all authors
Reviewers' Comments to Author
Reviewer 2
Firstly, we would like to thank to the Reviewer 2 for your valuable comments and suggestions provided to our work.
We apologize to you for the language mistakes. The manuscript has been revised again by a native English speaker with wide experience in translations for the field of Sport Physiology and Movement Sciences
All comments have been considered point-by-point in the current version with the aim to improve the quality of the manuscript. We have also marked in red colour all changes done through the manuscript.
---
Comment 1
To see the point at which there is a drop in the respiratory exchange ratio (RER), it could be interesting to show evolution of V̇CO2, in figure 1
Response
Thanks for your suggestion. Let us clarify that Figure 1 includes the evolution of four dependent variables of the study (V̇O2, Vr, SF and SL) during the 800 m field test.
The evolution of V̇CO2 was not included in the Figure to avoid adding more misunderstanding to the interpretation of the key data for the purpose of our study. However, the V̇CO2 variable has been analysed in our study, considered together with V̇O2 to estimate the respiratory exchange ratio (RER) (V̇CO2/ V̇O2), and thus, we used the RER as a criterion for determining the transition from phase I to phase II (according to the criteria defined by Whipp and Rossiter (2005) or Whipp, Ward, Lamarra, Davis, & Wasserman (1982) See references 27 and 28).
In any case, and to respond the Reviewer 2, we include below two figures showing the evolution of V̇CO2 and RER during the test.
---
Comment 2
Correlation value of lactate is explained in abstract but later it is not present in results, it is necessary to explain.
Response (Line 337-339)
Thanks for your detailed review. As you mentioned, the correlation value of lactate was included in the abstract (Line 25-26) indicating: "Our results showed that only the peak of blood lactate concentration after the 800 m test was correlated with the race time (p = .047)".
However, and responding to your comment, we had included in the result´s section of our first version of the manuscript this pattern of data, as it was showed in lines 197- 200 (in the corrected version of the manuscript it appears in lines 337-339): "When analysing the relationship between all the different variables measured and the time obtained in the 800 m test, only the [La -]peak value showed a statistically significant negative correlation (r= -.714, p = .047)"
---
Comment 3
Authors comment in methods that Heart rate (HR) is measured, but there isn’t any reference about this variable in results or discussion, why results are not explained?
Response (Line 262-264)
Many thanks for your observation. According to your suggestion, we have added now a few explanatory lines in this regard in section 2.4 (Data Analysis, Line 262-264): “Heart rate (HR) was used as a measure of secondary confirmatory criterion to those proposed for the determination of phase I (cardio dynamic component)”, so it was not included neither in the results nor in the discussion sections, because it was not defined as a study variable.
Below, we show the evolution of the HR during the test:
---
Given that we have tried to reply to the comments and suggestions by yourself, we hope that you will now find the enclosed manuscript suitable for publication in Sports.
Yours sincerely,
Vicente Torres on behalf of all authors
